# (3α,5α)3-Hydroxypregnan-20-one (3α,5α-THP) Regulation of the HPA Axis in the Context of Different Stressors and Sex

**DOI:** 10.3390/biom12081134

**Published:** 2022-08-18

**Authors:** Giorgia Boero, Ryan E. Tyler, Todd K. O’Buckley, Irina Balan, Joyce Besheer, A. Leslie Morrow

**Affiliations:** 1Bowles Center for Alcohol Studies, School of Medicine, University of North Carolina at Chapel Hill, 3027 Thurston Bowles Building, CB 7178, Chapel Hill, NC 27599, USA; 2Department of Psychiatry, School of Medicine, University of North Carolina at Chapel Hill, Chapel Hill, NC 27599, USA; 3Department of Pharmacology, School of Medicine, University of North Carolina at Chapel Hill, Chapel Hill, NC 27599, USA

**Keywords:** neuro-steroids, HPA axis, restraint stress, forced swim stress, 3α,5α-THP, allopregnanolone, CRF, CRFR1, corticosterone, hypothalamus

## Abstract

Corticotropin-releasing factor (CRF) regulates the stress response in the hypothalamus and modulates neurotransmission across the brain through CRF receptors. Acute stress increases hypothalamic CRF and the GABAergic neurosteroid (3α,5α)3-hydroxypregnan-20-one (3α,5α-THP). We previously showed that 3α,5α-THP regulation of CRF is sex and brain region dependent. In this study, we investigated 3α,5α-THP regulation of stress-induced hypothalamic CRF, CRF receptor type 1 (CRFR1), CRF binding protein (CRFBP), pro-opiomelanocortin (POMC), and glucocorticoid receptor (GR) by western blot and circulating corticosterone (CORT) by enzyme-linked immunosorbent assay (ELISA) in male and female Sprague Dawley rats. Tissue was collected after rats were injected with 3α,5α-THP (15 mg/kg, IP) or vehicle 15 min prior to 30 min of restraint stress (RS), or 10 min of forced swim stress (FSS) and 20 min recovery. The initial exposure to a stress stimulus increased circulating CORT levels in both males and females, but 3α,5α-THP attenuated the CORT response only in females after RS. 3α,5α-THP reduced GR levels in male and females, but differently between stressors. 3α,5α-THP decreased the CRF stress response after FSS in males and females, but after RS, only in female rats. 3α,5α-THP reduced the CRFR1, CRFBP, and POMC increases after RS and FSS in males, but in females only after FSS. Our results showed different stress responses following different types of stressors: 3α,5α-THP regulated the HPA axis at different levels, depending on sex.

## 1. Introduction

The activation of the hypothalamic-pituitary-adrenal (HPA) axis in response to stress stimuli is crucial for an organism. The stress response cascade is essential for the survival and dysregulation of HPA axis activity, and has been observed in several neuropsychiatric disorders, including drug addiction [1]. Corticotropin-release factor (CRF) is the main activator of the stress response. Identified by Vale in 1981 [2], CRF is a 41 amino-acid peptide that is well preserved across species. Released from the paraventricular nucleus (PVN) of hypothalamus, CRF reaches the anterior pituitary through the portal vessels and stimulates the transcription of the cDNA for pro-opiomelanacortin (POMC), and the subsequent synthesis and release of the adrenocorticotrophic hormone (ACTH) and β-endorphins [3] to the bloodstream. From the anterior pituitary, ACTH travels in the systemic circulation and reaches the zona fasciculata of the adrenal gland cortex, activating the synthesis and release of corticosteroids, mainly glucocorticoids and mineral-corticoids, such as corticosterone in rats (CORT) or cortisol in humans. Once corticosterone is released into the bloodstream, it is able to regulate its own production through the activation of glucocorticoid (GR) and mineral-corticoid receptors (MR) in different brain areas, including the hypothalamus [4]. The binding with GRs and MRs leads to negative feedback that decreases the production of CRF, then ACTH and finally corticosterone, ending the response to stress and bringing the organism back to homeostasis. Aberrant regulation of CRF signaling contributes to several neuropsychiatric disorders, including depression and anxiety [5].

The stress response can be regulated at many levels, not only by corticosteroids. In fact, CRF can regulate its own production. Once released from the hypothalamus, CRF travels in the blood, carried by the CRF binding-protein (CRFBP). CRFBP is a glycoprotein that plays a key role in the stress response. CRFBP binds CRF with high affinity and prevents the binding of CRF to its receptors, limiting the free availability and actions of CRF [6]. For this reason, CRFBP is generally considered as a CRF antagonist [7]. CRF exerts its biological effects through two types of post-synaptic membrane receptors, Gs-proteins coupled: CRF receptor type 1 (CRFR1) and CRF receptor type 2 (CRFR2). The two receptors share 70% of amino acid sequence [8], but they differ from each other based on localization and affinity. CRFR1 is present in hypothalamus, anterior pituitary, and other brain regions, while CRFR2 has been localized in brain and peripheral tissues [9]. CRFR1 has higher affinity for CRF than CRFR2 [10] and, because of its localization, CRFR1 appears to be involved in the regulation of HPA axis activity and the stress response [11].

HPA axis activation in response to stress can also be regulated through the GABAergic system. In fact, many regions regulating HPA axis function involve GABAergic connections: in the PVN of hypothalamus there is a high density of GABAergic afferents and one third of CRF neurons are controlled by GABAergic inputs [12]. Pharmacological manipulations have shown that GABAergic inhibition plays a key role in the regulation of HPA axis activity [13].

The activation of the HPA axis stress response leads to the synthesis and release of various neuroactive steroids, a class of metabolites of steroid hormones, that act as positive or negative allosteric modulators of different receptors, including glutamate and GABA_A_ receptors (GABA_A_R). (3α,5α)3-hydroxypregnan-20-one (3α,5α-THP, allopregnanolone) is the most potent positive modulator of the GABA_A_ receptors. It has been demonstrated that 3α,5α-THP increases following stress in animal models, in both plasma and brain [14,15], and in human serum following CRF or ACTH stimulation [16]. Previous studies have demonstrated that pre-treatment with 3α,5α-THP, and other neurosteroids such as 3α,5α-THDOC (another positive allosteric modulator of GABA_A_Rs) or progesterone attenuates stress-induced circulating ACTH and corticosterone [17,18]. Moreover, 3α,5α-THP treatment decreased hypothalamic CRF mRNA expression in male rats [19]. However, these studies did not consider sex, factors such as the type of stressor, or other levels of regulation of the stress axis.

We previously demonstrated that 3α,5α-THP regulation of basal levels of CRF is both sex and brain region dependent [20]. In the present study, we tested the ability of 3α,5α-THP to regulate key components of the stress response, challenging the HPA axis with two different stressors, restraint stress or forced swim stress in male and female Sprague-Dawley rats. We studied restraint stress, where the physical discomfort is secondary to the primary cognitive and psychological stress induced [21], vs. forced swim stress, where physical activity is required to survive. We analyzed key-components of the HPA axis stress response, including serum corticosterone levels and hypothalamic GRs, CRF, CRFR1, CRFBP, and POMC expression, in male and female rats. Our results show that 3α,5α-THP regulates the HPA axis stress response at different levels, depending on the type of stressor and the sex of the animal.

## 2. Materials and Methods

### 2.1. Animals

Male and female (>PN80-100) Sprague-Dawley rats bred in-house from Envigo stock were used in all experiments. Animals were pair-housed in a temperature and humidity controlled 12 h light/dark facility with ad libitum access to food and water. A total number of 77 male and 80 female Sprague-Dawley rats were used to perform the experiments in this study. Given that the main progesterone peak in the rat estrous cycle occurs in the early evening of proestrus and returns to basal levels by the morning of estrous [22,23], to avoid fluctuations in endogenous 3α,5α-THP due by estrus cycle all experiments were conducted between 8:00–11:30 a.m. All procedures were performed in accordance with guidelines approved by the Institutional Animal Care and Use Committee at the University of North Carolina at Chapel Hill (IACUC approval number: 21-118).

### 2.2. 3α,5α-THP Administration

3α,5α-THP (Steraloids Inc., Newport, RT, USA, #P3800) was dissolved in hydroxypropyl-β-cyclodextrin (45% *w*/*v* in water) at a concentration of 7.5 mg/mL. Solutions were prepared the day before the experiment and kept stirring at 4 °C overnight. Animals were randomly divided in two groups and, according to their weight, all rats received an IP injection of 3α,5α-THP (15 mg/kg) or an equivalent volume of vehicle (VEH). This dose was chosen based on our previous data, showing inhibition of CRF expression in male rats in specific brain regions [20,24]. This dose is known to have an anxiolytic and anti-convulsant effect, [25,26], but no hypnotic effect [27]. 15 min after VEH or 3α,5α-THP administration, rats were submitted to (a) restraint stress or (b) forced swim stress. 45 min after 3α,5α-THP administration, rats were euthanized by decapitation, brains and blood were collected, and brains were immediately frozen at −80 °C until the assay. Data from VEH groups are described as the baseline measure in experiments.

### 2.3. Stress Paradigms

#### 2.3.1. Restraint Stress

Animals were randomly assigned to control (no restraint stress, NRS) and stressed (restraint stress, RS) group, then 15 min after VEH or 3α,5α-THP administration rats were placed in plastic DecapiCone tubes (Braintree Scientific, Inc., Braintree, MA, USA, #DCL120) and subjected to restraint stress for 30 min in another room. The animals were secured inside the cones at the tail. A hole in the head of the cones allowed rats to breathe. Animals were not able to move or turn around. During the restraint, the animals had no access to food and water. Non-restraint rats (control group) remained in the home cages until behavioral experiments started. Rats were euthanized by decapitation immediately after the end of restraint stress, correspondent to 45 min after the VEH or 3α,5α-THP administration.

#### 2.3.2. Forced Swim Stress

The forced swim test was performed in a clear plastic cylinder (45.7 cm H × 20 cm diameter; Stoelting Co., 620 Wheat Ln, Wood Dale, IL, USA, #60160), filled with water (25 ±  1  °C) depth ~30 cm, so that rats could not touch the bottom of the cylinder with the tail. Animals were randomly assigned to control (no forced swim stress, NFSS) and stressed (forced swim stress, FSS) group. 15 min after VEH or 3α,5α-THP administration, rats were placed into the cylinder filled with water at the appropriate temperature and allowed to explore the environment. The test lasted for 10 min, then the animals were dried with absorbent towels and returned in the cage for 20 min. Cylinders were cleaned and water was changed after each animal. Rats were euthanized by decapitation 45 min after the VEH or 3α,5α-THP administration.

### 2.4. Immunoblotting

The brains were dissected using a brain block and the hypothalamus was homogenized and sonicated in ice-cold CelLytic MT lysis buffer (Sigma-Aldrich, Saint Louis, MO, USA, #C3228) with 1× HALT protease and phosphatase inhibitor (Thermo Fisher Scientific, 81 Wyman Street Waltham, MA, USA, #1861281). The tissue was left on ice for 30 min and then centrifugated at 14,000× *g* for 30 min at 4 °C. The supernatant was transferred to new tubes, used immediately or stored at −80 °C. Total protein was determined by the bicinchoninic acid assay (BCA, Thermo Fisher Scientific #23228, #1859078). The proteins (40 μg/lane) were denatured at 95 °C for 5 min in LDS sample buffer (Thermo Fisher Scientific #NP0007) and sample reducing agent (Thermo Fisher Scientific #NP0009) and were resolved by NuPAGE™ 10% Bis-Tris Midi Protein Gel (Thermo Fisher Scientific #WG1202, #WG1203) electrophoresis at 125 V for 10 min and 165 V for the rest of the running and transferred to polyvinylidene fluoride membranes (iBlot2 PVDF regular stacks, Thermo Fisher Scientific #IB24001) using the iBlot 2 Dry Blotting System (Thermo Fisher Scientific).

The blots were blocked with 5% Blotting-Grade Blocker (Bio-Rad Laboratories, 1000 Alfred Nobel Drive, Hercules, CA, USA, #1706404) in PBS-T (0.5% Tween-20) for 2 h (room temperature) and exposed to primary antibody overnight (4 °C), followed by horseradish peroxidase-labeled secondary antibodies for 1 h (room temperature). PBS-T (0.5%) was used to wash the blot 3 times (15 min each, room temperature) after incubation with primary and secondary antibodies. Blots were stripped with a commercial stripping solution (Restore Western Blot stripping buffer, Thermo Fisher Scientific #21059), washed, and blocked again for 1 h, before being re-probed with different primary antibodies (3–5 times).

Immunoreactive bands were visualized with Western Lightning Plus (Perkin Elmer, 940 Winter Street, Waltham, MA, USA, #NEL105001EA) followed by detection with enhanced chemiluminescence (ImageQuant LAS4000, Cytiva, 100 Results Wy, Marlborough, MA, USA,). Bands were analyzed using ImageQuant TL v8.1.0.0 software (Cytiva, 100 Results Wy, Marlborough, MA, USA). Each densitometric measurement was divided by the corresponding β-actin densitometric measurement and the results are expressed as percentage versus control group (VEH, Non-stressed).

### 2.5. Antibodies

Antibodies were commercially obtained and used according to the manufacturer’s instructions. Primary antibodies are listed in Appendix A. Horseradish peroxidase-labeled secondary antibodies were anti-rabbit (Cell Signaling Technology, 3 Trask Ln, Danvers, MA, USA, #7074), anti-mouse (Cell Signaling Technology #7076), and anti-goat IgG (Thermo Fisher Scientific #A24452).

### 2.6. Corticosterone Assay

After decapitation, blood was collected in glass tubes (BD, 1 Becton Drive Franklin Lakes, NJ, USA, BD Vacutainer^®^ Serum tubes #366430), kept on ice until the end of the experiment and immediately centrifugated at 1750× *g* for 15 min at 4 °C. Serum was transferred to Eppendorf Flex 2 mL tubes and frozen at −80 °C until the assay. Serum was assayed with ELISA kit for corticosterone (MP Biomedicals, 9 Goddard Irvine, CA, USA, #07DE-9922) following the manufacturer’s instructions. Results are expressed as nanograms/milliliter (ng/mL).

### 2.7. Statistical Analysis

All immunoblotting results are presented as % change vs VEH no stress ± SEM. Corticosterone data are presented as mean ± SEM in ng/mL. Statistical analyses were conducted by two-way ANOVA and significant interactions were followed up by Tukey Honest Significant Differences (HSD) test. All data were analyzed with GraphPad Prism 9 (GraphPad Software, 2236 Avenida De La Playa La Jolla, CA, USA). A value of *p* < 0.05 was considered statistically significant.

## 3. Results

### 3.1. 3α,5α-THP Partially Prevents the Corticosterone Stress Response after Restraint Stress in Female, but Not in Male Rats

As corticosterone (CORT) is a common parameter to evaluate the stress response, we measured serum levels after restraint stress or forced swim stress, in male and female rats. As expected, circulating CORT levels increased in males and females after restraint stress or forced swim stress (Figure 1). Restraint stress increased CORT levels by 595% in male and 295% in female rats, respectively (Males restraint stress: Treatment F(1,43) = 0.62, *p* = 0.44; Stress F(2,43) = 35.34, *p* < 0.0001; Interaction F(2,43) = 0.1, *p* = 0.904; Females restraint stress: Treatment F(1,27) = 8.6, *p* = 0.0068; Stress F(1,27) = 80.88, *p* < 0.0001; Interaction between factors F(1,27) = 5.643, *p* = 0.025). Forced swim stress increased CORT levels by 266% in male and 321% in female rats, respectively. (Males forced swim stress: Treatment F(1,33) = 3.15, *p* = 0.09; Stress F(1,33) = 98.36, *p* < 0.0001; Interaction between factors F(1,33) = 1.887, *p* = 0.999; Females forced swim stress: Treatment F(1,29) = 11.64, *p* = 0.0019; Stress F(1,29) = 82.97, *p* < 0.0001; Interaction between factors F(1,29) = 0.0023, *p* = 0.962). Consistent with previous literature, 3α,5α-THP had no effect on circulating CORT levels in the absence of either stress exposure. Surprising, 3α,5α-THP failed to attenuate the stress-induced CORT escalation following either restraint stress or forced swim stress in male rats. However, in female rats, 3α,5α-THP administration slightly decreased the CORT stress response following restraint stress, but not following forced swim stress.

### 3.2. Glucocorticoid Receptor (GR) Regulation by 3α,5α-THP Differs Based on Type of Stressor, but Not Sex

Given 3α,5α-THP did not affect CORT circulating levels, we investigated the possibility that 3α,5α-THP exerted its action at a different level of the stress response in the hypothalamus. Through a negative feedback loop, CORT is able to regulate its own production binding GRs in different brain regions, including the hypothalamus [4]. Our results show that stress and 3α,5α-THP regulation of GR in the hypothalamus differ based on the stress stimuli.

In both male and female animals, restraint stress increased hypothalamic GR expression (Figure 2). GR expression was increased by 29% in males and 41% in females. 3α,5α-THP reduced hypothalamic GR expression following restraint stress by 37% in males, but not in females. (Males restraint stress: Treatment F(1,30) = 63.12, *p* < 0.0001; Stress F(1,30) = 5.305, *p* = 0.028; Interaction between factors F(1,30) = 4.577, *p* = 0.0407. Females restraint stress: Treatment F(1,29) = 7.554, *p* = 0.0102; Stress F(1,29) = 20.985, *p* < 0.0001; Interaction between factors F(1,29) = 0.9638, *p* = 0.3343). In contrast, forced swim stress did not induce any changes in GR levels in either males or females. However, 3α,5α-THP reduced hypothalamic GR expression in the absence of forced swim stress in males (19%), and both with (52%) and without (47%) forced swim stress in females. This main effect is not sex dependent (Males forced swim stress: Treatment F(1,35) = 10.10, *p* = 0.0031; Stress F(1,35) = 0.458, *p* = 0.503; Interaction between factors F(1,35) = 0.626, *p* = 0.4342; Females forced swim stress: Treatment F(1,35) = 70.58, *p* < 0.0001; Stress F(1,35) = 0.1778, *p* = 0.676; Interaction between factors F(1,35) = 0.1778, *p* = 0.676).

### 3.3. 3α,5α-THP Diminished CRF Stress-Induced Response after Forced Swim Stress, but Only in Female Rats after Restraint Stress

It is well known that stress stimulus increases in hypothalamic CRF levels, leading a cascade of events called stress response. To confirm the activation of the HPA axis, we measured CRF expression in male and female rats, following restraint or forced swim stress.

As expected, CRF levels increased in the hypothalamus of male and female rats after restraint stress or forced swim stress (Figure 3). Restraint stress increased hypothalamic CRF by 76% in males and by 144% in females (Males restraint stress: Treatment F(1,31) = 3.825, *p* = 0.0595; Stress F(1,31) = 123.2, *p* < 0.0001; Interaction between factors F(1,31) = 20.47, *p* < 0.0001; Females restraint stress: Treatment F(1,30) = 8.706, *p* = 0.0061; Stress F(1,30) = 59.04, *p* < 0.0001; Interaction between factors F(1,30) = 24.23, *p* < 0.0001). Moreover, forced swim stress increased CRF expression by 55% in males and 132% in females (Males forced swim stress: Treatment F(1,33) = 33.63, *p* < 0.0001; Stress F(1,32) = 13.60, *p* = 0.0008; Interaction between factors F(1,32) = 8.778, *p* = 0.0057. Females forced swim stress: Treatment F(1,36) = 49.80, *p* < 0.0001; Stress F(1,36) = 15.54, *p* = 0.0004; Interaction between factors F(1,36) = 28.55, *p* < 0.0001). 3α,5α-THP reduced the CRF stress-induced response in female animals, following both restraint and forced swim stress (Figure 3). In male rats, 3α,5α-THP reduced CRF stress-induced rise after forced swim stress, but 3α,5α-THP failed to decrease the CRF response following restraint stress. Surprisingly, following restraint stress, 3α,5α-THP resulted in a synergic effect with the stress and increased the CRF stress response in male rats.

### 3.4. 3α,5α-THP Regulation of CRFR1 Is Both Stressor and Sex-Dependent

Similar to CORT, CRF regulates its own production via binding of its receptors (CRFR1/CRFR2) at the hypothalamic level. CRFR1 mediates the activation of the HPA axis [11,28] and its activation induces anxiety-like behavior [29]. To understand 3α,5α-THP effects on the stress response, we measured CRFR1 expression in the hypothalamus of male and female rats, following restraint or forced swim stress.

CRFR1 response to stress differed between stressors, in both male and female rats. In male animals, restraint stress induced an increase by 60% in hypothalamic CRFR1 expression and 3α,5α-THP prevented CRFR1 rise after stress (Figure 4. Males restraint stress: Treatment F(1,28) = 13.69, *p* = 0.0009; Stress F(1,28) = 5.782, *p* = 0.0230; Interaction between factors F(1,28) = 9.413, *p* = 0.0047). In contrast, in female rats restraint stress did affect hypothalamic CRFR1 and 3α,5α-THP did not alter CRFR1 baseline levels (Females restraint stress: Treatment F(1,27) = 0.2101, *p* = 0.6504; Stress F(1,27) = 0.2035, *p* = 0.6555; Interaction between factors F(1,27) = 0.2045, *p* = 0.6548). Forced swim stress did not alter CRFR1 levels in male rats; however, 3α,5α-THP treatment reduced CRFR1 levels by 17% in male rats following forced swim stress (Males forced swim stress: Treatment F(1,32) = 4.336, *p* = 0.0454; Stress (1,32) = 0.1472, *p* = 0.7037; Interaction between factors F(1,32) = 8.480, *p* = 0.0065). As observed in males, forced swim stress did not affect CRFR1 levels in female animals. 3α,5α-THP induced a decrease in hypothalamic baseline CRFR1 in both with and without stress group by 44% and 39%, respectively (Females forced swim stress: Treatment F(1,34) = 47.15, *p* < 0.0001; Stress F(1,34) = 1.076, *p* = 0.3070; Interaction between factors F(1,34) = 5.350, *p* = 0.0269).

### 3.5. 3α,5α-THP Effect on CRF Binding Protein (CRFBP) Differs between Stressors and Sex

CRF effects are mediated and controlled not only by CRF receptors, but also by the CRF-binding protein (CRFBP). CRFBP binds CRF with high affinity and sequesters it away from the receptors, reducing CRF activity [30]. Given CRFBP is an important component of the stress response, we verified the possibility that 3α,5α-THP regulates CRFBP levels in male and female rats, following restraint stress or forced swim stress.

In male rats, restraint stress per se reduced hypothalamic CRFBP by 29%. In addition, 3α,5α-THP treatment did not change the reduction induced by restraint stress on CRFBP levels in the hypothalamus of male rats (−27%). (Figure 5. Males restraint stress: Treatment F(1,32) = 8.801, *p* = 0.0057; Stress F(1,32) = 17.03, *p* = 0.0002; Interaction between factors F(1,32) = 1.894, *p* = 0.1784). In female rats, we did not detect any change in hypothalamic CRFBP following restraint stress (Females restraint stress: Treatment F(1,30) = 3.109, *p* = 0.0880; Stress F(1,30) = 3.313, *p* = 0.0787; Interaction between factors F(1,30) = 2.001, *p* = 0.1675). In contrast, forced swim stress increased hypothalamic CRFBP by 29% in male rats. 3α,5α-THP administration reduced the CRFBP levels swim stress-induced (Males forced swim stress: Treatment F(1,33) = 7.166, *p* = 0.0115; Stress (1,33) = 6.183, *p* = 0.0181; Interaction between factors F(1,33) = 6.450, *p* = 0.0160). In female animals, forced swim stress did not affect CRFBP levels in hypothalamus. 3α,5α-THP decreased CRFBP levels in female rats, but this effect is not related to the swim stress exposure (Females forced swim stress: Treatment F(1,35) = 37.00, *p* < 0.0001; Stress F(1,35) = 1.397, *p* = 0.2452; Interaction between factors F(1,35) = 3.358, *p* = 0.0754).

### 3.6. 3α,5α-THP Decreased Stress-Induced Hypothalamic POMC Levels in Male, but Not in Female Rats

CRF stimulates the synthesis and release of POMC, precursor of ACTH, from the anterior pituitary. Although POMC exerts its main effect in the pituitary, the POMC gene is found in different brain regions, including hypothalamus [31]. We explored the possibility that 3α,5α-THP regulates POMC levels, in male and female rats, following restraint stress or forced swim stress.

In male rats, hypothalamic POMC levels increased following restraint stress by 35%. 3α,5α-THP attenuated the stress response, reducing POMC expression after restraint stress (Figure 6. Males restraint stress: Treatment F(1,32) = 23.17, *p* < 0.0001; Stress F(1,32) = 7.480, *p* = 0.0101; Interaction between factors F(1,32) = 7.483, *p* = 0.0101). In female rats, we observed a general increase of POMC levels following restraint stress; the treatment with 3α,5α-THP failed to attenuate the effect of stress (Females restraint stress: Treatment F(1,28) = 2.321, *p* = 0.1389; Stress F(1,28) = 27.34, *p* < 0.0001; Interaction between factors F(1,28) = 0.00243, *p* = 0.9611). As previously described for the restraint stress, forced swim stress increased POMC expression by 25% in male rats and 3α,5α-THP inhibited the rise in POMC levels following forced swim stress (Males forced swim stress: Treatment F(1,33) = 21.46, *p* < 0.0001; Stress (1,33) = 4.895, *p* = 0.0340; Interaction between factors F(1,33) = 7.399, *p* = 0.0103). Forced swim stress did not affect hypothalamic POMC levels in female animals. Nevertheless, in female rats, 3α,5α-THP treatment decreased hypothalamic POMC levels (Females forced swim stress: Treatment F(1,35) = 9.564, *p* = 0.0039; Stress F(1,35) = 0.16, *p* = 0.6916; Interaction between factors F(1,35) = 10.83, *p* = 0.0023).

## 4. Discussion

We previously discovered that 3α,5α-THP regulation of CRF expression is sex and region-specific at baseline levels [20]. The aim of this work was to investigate the effect of 3α,5α-THP on the HPA axis stress response following two different types of stressors: a psychological stress, such as restraint stress, and a physical stress, such as forced swim stress. Our results show different stress responses following different type of stressors and that 3α,5α-THP regulates the HPA axis at various different levels, depending on sex (see Table 1 for a summary of the results).

### 4.1. Circulating Corticosterone

Exposure to acute stress rapidly increases 3α,5α-THP levels in both plasma and brain [14,15]. Acute stress-induced increase of 3α,5α-THP is considered part of the HPA axis negative feedback mechanism, to promote homeostasis by reducing HPA axis activation due to 3α,5α-THP action on GABAergic neurotransmission [32,33]. Additionally, 3α,5α-THP influences the genomic pathway by downregulating gene expression of key-components of HPA axis stress response. In fact, it has been reported that 3α,5α-THP (and other GABAergic neuro-steroids) regulate HPA axis activation by reducing corticosterone levels, CRF mRNA transcript and/or ACTH levels in male rodents [13,17,18,19].

As expected, our results showed that serum corticosterone levels increased in both male and female rats, following restraint stress or forced swim stress. Surprisingly, 3α,5α-THP failed to prevent the corticosterone stress response, in male animals, regardless of the type of stressor. This paradoxical effect is quite surprising and cannot be explained by sex or type of stressor, since we evaluated the same stressors as previously reported in male rats. In our previous work [20], we showed that 3α,5α-THP levels were increased in both serum and brain regions, including hypothalamus, at the same time point as the present study (45 min after the initial 3α,5α-THP injection). For this reason, we assumed that the dose and timing used in this study were appropriate to detect an effect on the stress response. However, given the different duration of the stress exposure (30 min for the restraint stress vs. 10 min for the forced swim stress) and the recovery time (no recovery time after restraint stress vs. 20 min recovery after forced swim stress), we cannot exclude the possibility that differences between stressors were influenced by those factors. Future studies are needed to clarify this point.

The contrasting findings between our results and previous data that demonstrated the ability of GABAergic neuro-steroids to inhibit the corticosterone response to stress are not unique [34]. For example, other studies showed that administration of 3α,5α-THP can exacerbate the corticosterone response following acute stress in mice [35]. We speculate that the differences in 3α,5α-THP effects on corticosterone levels could be due to differences in the 3α,5α-THP dose administered or the timing chosen to collect the samples (e.g., plasma, brain tissues, etc.). In fact, the protocols used in previous studies employed different stressors, 3α,5α-THP doses, and collection times. For example, Patchev et al., 1994 [19] activated the HPA axis of Wistar rats with I.C.V. administration of CRF 0.5 μg/rat following subcutaneous administration of 1 mg of 3α,5α-THP; Patchev et al., 1996 [18] injected Wistar rats I.P. with a dose of 50 μg/kg of 3α,5α-THP and 30 min later exposed the animals to air-puff stressor; Sarkar et al., 2011 [13] used C57BL/6 or Gabrd−/− mice and direct injected THDOC into the PVN. However, it is also possible that unknown variables contribute to the lack of replication of previous reports.

In contrast, 3α,5α-THP treatment prior to restraint stress partially reduced the stress-induced corticosterone response in the females. Because this sex difference was unique to the restraint stress procedure, it is unlikely to be explained by estrus cycle effects, but rather may involve effects of psychological stress that are unique to females. Upon further evaluation, we found circulating corticosterone levels under control conditions were higher in female than male rats, consistent with previous data that showed female rats in resting conditions displayed higher serum corticosterone concentrations [36,37], depending of the stage of the estrous cycle [38]. We did not monitor the estrous phase in the female rats used for this study, to avoid the increase in corticosterone and 3α,5α-THP levels due to the stress of the monitoring procedure. Indeed, the female animals used were unlikely to have bee sampled during the estrus phase, based on their elevated corticosterone levels, which are known to be equivalent to males during this phase [38]. Nevertheless, we cannot completely exclude a potential role of the estrous cycle in our results.

Consistent with previous data, 3α,5α-THP did not affect serum corticosterone levels in resting conditions. Following restraint stress, serum corticosterone levels increased in both male and female rats; in male rats this increase was greater than in females, probably due to their lower resting corticosterone levels. Similarly, exposure to forced swim stress increased circulating corticosterone levels to a similar magnitude in both male and female groups. Interestingly, the magnitude of the increase in corticosterone following forced swim stress exposure is similar between males and females, but male rats displayed a lower percentage of increase following swim stress than restraint stress. This difference between the two types of stressors has also been observed in mice [39], confirming a different activation of the HPA axis following different stress exposures. Furthermore, 3α,5α-THP failed to attenuate the corticosterone stress response following forced swim stress in both males and females.

### 4.2. Hypothalamic CRF

Despite the lack of effect on corticosterone levels, the final product of the stress response, our data showed that 3α,5α-THP regulated other important markers of the stress response in the hypothalamus in a sex- and stress stimulus-dependent manner. Restraint stress affected several hypothalamic components of the stress response in a different manner than forced swim stress. For example, exposure to restraint stress activated the HPA axis by increasing the expression of GRs, CRF, and POMC in male animals. In contrast, forced swim stress did not induce changes in GRs and CRFR1 in male or female rats, or alter CRFBP or POMC levels in female rats.

In our previous study [20] we showed a sexual dimorphism in 3α,5α-THP regulation of hypothalamic CRF signals under resting conditions. Consistent with previous data [19], we observed that 3α,5α-THP had no effect in female rats but significantly decreased CRF mRNA (but not peptide expression) in male rats. These new data support our previous observation, confirming a different 3α,5α-THP regulation of HPA axis depend upon sex. As expected, the exposure to both restraint stress and forced swim stress raised hypothalamic CRF content. Despite the similar stress response, 3α,5α-THP reduced the stress-induced increase in hypothalamic CRF following swim stress in both male and female animals. Moreover, 3α,5α-THP decreased the CRF stress response following restraint stress in female rats. However, this effect was not observed after restraint stress in male rats, where 3α,5α-THP failed to attenuate the CRF stress increase, but instead enhanced CRF. These results suggest differences in the regulation of HPA axis by 3α,5α-THP depending on the type of stressor, but this effect is also sex-specific.

### 4.3. CRF Binding Protein (CRFBP)

CRF activity is correlated with the concentration of the binding protein, CRFBP, the carrier that transports CRF through the vessels and sequesters it away from the receptors [6]. Our data showed that both sex and the type of stress influenced hypothalamic CRFBP expression. While restraint stress reduced CRFBP, forced swim stress increased the levels of this protein in male rats. These diverse effects suggest a stress-stimulus-dependent HPA axis activation; 3α,5α-THP did not alter the reduced CRFBP levels in male rats following restraint stress. This result could be linked to the higher CRF levels in males following restraint stress since the stress-induced decrease in CRFBP levels could lead to an increase of CRF availability and activity. In contrast, following forced swim stress, 3α,5α-THP attenuated the CRFBP stress-induced response, in male rats. These results suggest, once again, that 3α,5α-THP action is dependent on the type of stress.

Confirming the sex differences in the activation and regulation of HPA axis, in female rats we did not detect any change in CRFBP levels following restraint stress or forced swim stress. Further studies are necessary to clarify the role of CRFBP in the female stress response.

### 4.4. CRF Receptor 1 (CRFR1)

Another important component of the stress response is represented by the CRF receptors. In this study we analyzed only the CRFR1 type, that exhibit ubiquitous distribution in the brain. CRFR1 has been well studied in neuropsychiatric disorders such depression, anxiety, and addiction in the attempt to reduce the overexpression of CRF and the hyperactivity of HPA axis observed in these disorders [40]. Preclinical studies showed positive results using a CRFR1 antagonist [41,42], but unfortunately in human studies this approach did not appear to have efficacy as a monotherapy for neuropsychiatric disorders [43]. Our data showed that hypothalamic CRFR1 expression is influenced by the type of stress in male animals. While restraint stress induced an increase in CRFR1 levels, forced swim stress did not modify CRF receptors. However, 3α,5α-THP decreased CRFR1 expression following both restraint stress and forced swim stress in male rats. It is possible that the lack of 3α,5α-THP effect on CRF levels observed in male rats is compensated by the decrease in stress-induced CRFR1 following restraint stress. This reduction in CRF receptors could be a 3α,5α-THP-mediated indirect mechanism to reduce CRF signaling; although CRF levels are still high, the decrease in the abundance of CRFR1 leads to a decrease in CRF signaling.

It is very interesting that neither restraint stress nor forced swim stress altered CRFR1 expression in female animals, and 3α,5α-THP did not affect CRFR1 levels following restraint stress. It appears that the absence of 3α,5α-THP effect following restraint stress is related to the direct effect of 3α,5α-THP on CRF expression in the absence of stress in female rats. In contrast, 3α,5α-THP reduced CRFR1 concentration in the absence of and following forced swim stress. This discrepancy is clearly attributable to the type of stress.

### 4.5. Glucocorticoid Receptors (GRs)

It has been demonstrated that GRs have a dual mechanism of action in the regulation of the stress response: a rapid regulation lead by membrane receptors and a genomic mechanism lead by nuclear receptors [44]. In this work, we measured the total fraction of hypothalamic GRs, so we were not able to discriminate between membrane and nuclear receptors. However, our data support the theory that the increase in hypothalamic GR-mediated signaling is an essential mechanism for the activation of the stress response. The 3α,5α-THP treatment prevented the stress-induced increase in GR expression following restraint stress in male rats. Reducing the binding substrate of corticosterone, it is likely that 3α,5α-THP is indirectly reducing corticosterone effects. Interestingly, the decrease induced by 3α,5α-THP in males was higher than females following restraint stress; this effect could be related to the higher corticosterone response observed in the males vs. the females. While restraint stress induced an increase in GRs levels in males, forced swim stress did not affect hypothalamic GR levels in both male and female animals. The 3α,5α-THP-induced downregulation in GRs observed in male rats, following restraint stress, may represent an indirect mechanism to reduce corticosterone action through its receptors. Given that corticosterone binds both GRs and MRs, further studies need to determine the effect of 3α,5α-THP on MRs.

The changes observed in GR expression (and CRFR1) after 3α,5α-THP administration, suggest that 3α,5α-THP may have a genomic mechanism that influences receptor expression, independent from the well-known 3α,5α-THP GABAergic action. Even though there is no evidence that 3α,5α-THP directly binds or activates GRs [45], in vitro studies showed that 3α,5α-THP and other pregnenolone derivates inhibited, in a dose-dependent manner, GR-mediated gene transcription [46]. 3α,5α-THP might act at downstream levels, interacting with various protein kinases involved in the regulation of GR function, in a similar way as antidepressant drugs [47,48]. It has also been demonstrated in rats that the ability of 3α,5α-THP to prevent the effect of restraint stress was reduced by blocking GRs with RU486 (a progesterone/glucocorticoid receptor antagonist) [49]. Moreover, it has been shown that 3α,5α-THP, after conversion to 3α,5α-DHP, can activate progesterone receptor-mediated gene transcription [50]. Additionally, 3α,5α-THP is able to activate another nuclear receptor known as pregnane xenobiotic receptor (PXR) [51]. PXR regulates cholesterol metabolism through the cholesterol-binding translocator protein (TSPO), essential for the transport of cholesterol into the mitochondria, where steroidogenesis starts, suggesting that 3α,5α-THP can also regulate its own production [52,53]. Previous studies demonstrated that the activation of PXR increased corticosterone levels with no effect on ACTH secretion [54]. Furthermore, a reciprocal regulation between nuclear receptors (such as nurr1 and nurr77) and CRF signaling was found [55]. Finally, 3α,5α-THP may act on other nuclear receptors, regulating the CRFR1 or GR-mediated gene transcription involved in the stress response. Further studies are needed to address these possibilities.

### 4.6. Pro-Opiomelanocortin (POMC)

Increasing CRF following a stress stimulus leads to the synthesis and release of ACTH from the pituitary. POMC is the precursor of ACTH, an essential hormone in the cascade of stress response. Our data showed a strong sex difference in the regulation of this precursor, depending also on the type of stressor. In male rats, both restraint and forced swim stress induced an increase in hypothalamic POMC levels, and 3α,5α-THP attenuated the stress-induced response in male rats, following both type of stressors. Restraint stress also increased POMC levels in female rats, but 3α,5α-THP failed to reduce this stress-dependent effect. In contrast, forced swim stress did not change hypothalamic POMC concentration in females. However, in female animals, 3α,5α-THP reduced POMC levels in absence of stress.

Despite the lack of 3α,5α-THP effect on corticosterone levels, our data demonstrated the ability of this neuro-steroid to modify other important components of the HPA axis following a stress exposure. These 3α,5α-THP effects are dependent on sex and the type of stressor the animals faced. Corticosterone and other glucocorticoids are well-known since the 1970s for their immunosuppressive and anti-inflammatory proprieties. However, in the past decades, several studies proved that glucocorticoids have also a pro-inflammatory impact on immune system [56,57]. Previous data from our lab [24,58] showed that 3α,5α-THP inhibits toll-like receptor pathways, reducing the production of pro-inflammatory chemokines and cytokines. Stress can activate the inflammatory response in the brain and peripherally [59]. Acute stress activates microglia and increases pro-inflammatory cytokine production [60], as observed in neuropsychiatric disorders such as depression [61], anxiety [62], schizophrenia [63], and autism spectrum disorders [64]. Finally, 3α,5α-THP might be controlling HPA axis stress-induced response through the immune system, reducing inflammation caused by stress. This idea will also require further study.

## 5. Conclusions

3α,5α-THP plays an essential regulatory role on the neuroendocrine modifications induced by the stress response. Our data show various different levels of HPA axis regulation by 3α,5α-THP, depending on the type of stressor and sex. These results provide a more complete understanding of HPA axis biomarkers involved in stress responses and the role that 3α,5α-THP plays in stress regulation. Furthermore, these data show the importance of sex as biological variable in the regulation of the HPA axis, given the differences in steroidogenesis and treatment responses previously observed [65]. Moreover, different types of stressors produce different activation of various HPA axis components, and, accordingly, different 3α,5α-THP effects. Further studies are required to determine the role of 3α,5α-THP in the regulation of aberrant HPA axis activity. The recent evidence in clinical studies, using the 3α,5α-THP formulation, brexanolone, and its FDA approval for the treatment of post-partum depression [66], underscore the therapeutic importance of neuro-steroid signaling in the brain [67]. Stimulation of 3α,5α-THP biosynthesis through development of selective neuro-steroidogenic drugs or precursor administration may also be effective for the treatment of stress-related disorders.

## Figures and Tables

**Figure 1 biomolecules-12-01134-f001:**
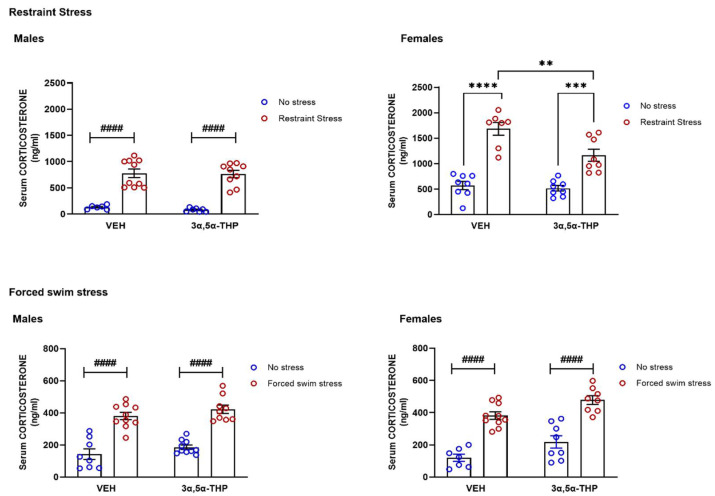
**3α,5α-THP partially reduced the restraint stress-induced corticosterone increased in female rats.** As expected, both acute restraint stress and forced swim stress induced an increase in CORT circulating levels, in male and female rats (Males VEH NRS: 130.9 ± 15.92 ng/mL vs. VEH RS: 779.1 ± 81.8 ng/mL, stress effect *p* < 0.0001; males VEH NFSS: 143.2 ± 32.8 ng/mL vs. VEH FSS: 380.5 ± 23.2 ng/mL, stress effect *p* < 0.0001. Females VEH NRS: 571.3 ± 80.5 ng/mL vs. VEH RS: 1684 ± 128.1 ng/mL, stress effect *p* < 0.0001; females VEH NFSS: 119.14 ± 22.2 ng/mL, vs. VEH FSS: 382 ± 23 ng/mL, stress effect *p* < 0.0001). Surprisingly, 45 min after the injection, 3α,5α-THP failed to attenuate the CORT stress response, in male and female rats, following restraint or forced swim stress (Males 3α,5α-THP NRS: 79.04 ± 13.1 ng/mL vs. 3α,5α-THP RS: 762.8.1 ± 70.32 ng/mL, stress effect *p* < 0.0001; males VEH RS: 779.1 ± 81.8 ng/mL vs. 3α,5α-THP RS: 762.8.1 ± 70.32 ng/mL, n.s.; males 3α,5α-THP NFSS: 185.7 ± 13.6 ng/mL vs. 3α,5α-THP FSS: 423 ± 25.8 ng/mL, stress effect *p* < 0.0001; male VEH FSS: 380.5 ± 23.2 ng/mL vs. 3α,5α-THP FSS: 423 ± 25.8 ng/mL, n.s. Females 3α,5α-THP NRS: 516.9 ± 54.3 ng/mL vs. 3α,5α-THP RS: 1164.6 ± 118 ng/mL, *p* < 0.001; females VEH RS: 1684 ± 128.1 ng/mL vs. 3α,5α-THP RS: 1164.6 ± 118 ng/mL, *p* < 001; females 3α,5α-THP NFSS: 218.5 ± 38.9 ng/mL vs. 3α,5α-THP FSS: 478.5 ± 27 ng/mL, stress effect *p* < 0.0001; females VEH FSS: 382 ± 23 ng/mL vs. 3α,5α-THP FSS: 478.5 ± 27 ng/mL, n.s.). Significant effects were found using Two-way ANOVA ^####^ *p* < 0.0001, or following Tukey HSD test **** *p* < 0.0001, *** *p* < 0.001, ** *p* < 0.01. Data are represented as mean ± SEM. Abbreviations: VEH = rats treated with vehicle; 3α,5α-THP = rats treated with 3α,5α-THP; NRS = non-restraint stress; RS = restraint stress; NFSS = non-forced swim stress; FSS = forced swim stress.

**Figure 2 biomolecules-12-01134-f002:**
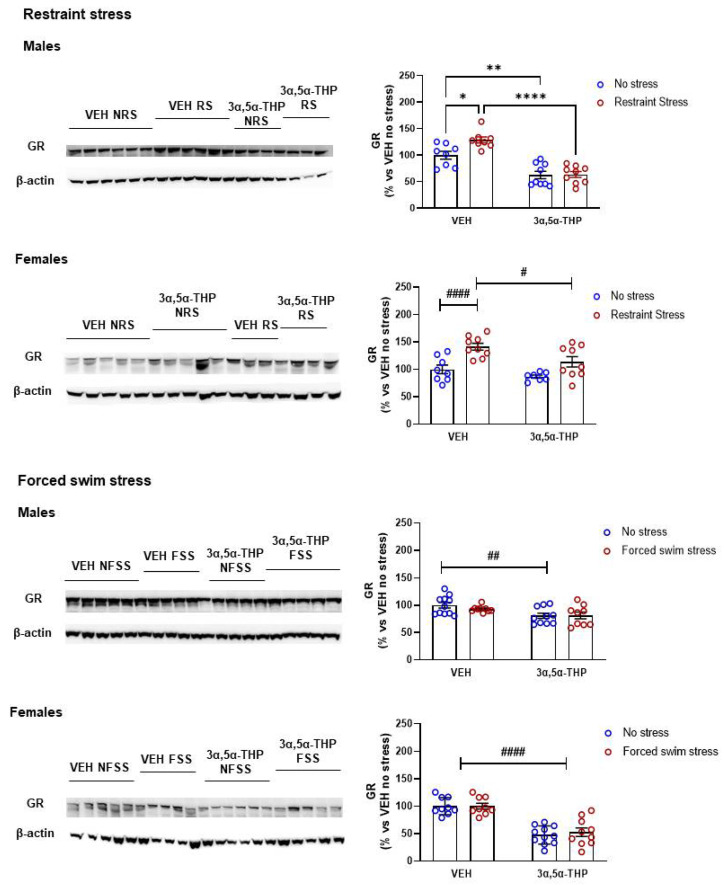
**3α,5α-THP reduced restraint stress-induced hypothalamic GRs increase in male rats.** Restraint stress, but not forced swim stress, induced an increase in hypothalamic GR expression, in both male and female rats (Males VEH RS: 129 ± 5.8% vs. VEH NRS, *p* < 0.05; Males VEH FSS: 93 ± 1.9% vs. VEH NFSS, n.s.. Females VEH RS: 41 ± 6.2% vs. VEH NRS, stress effect *p* < 0.0001; Females VEH FSS: 99.9 ± 5.2 vs. VEH NFSS, n.s.). 3α,5α-THP attenuated the restraint stress-induced enhancement in hypothalamic GR levels in both male and female rats (Males 3α,5α-THP NRS: 64 ± 5.5% vs. VEH NRS, *p* < 0.01; Males 3α,5α-THP RS: 63 ± 7% vs. VEH RS, *p* < 0.0001. Females 3α,5α-THP NRS: 87 ± 2.8% vs. VEH NRS, Females 3α,5α-THP RS: 114 ± 9.5% vs. VEH RS, treatment *p* < 0.05). 3α,5α-THP treatment resulted in a decrease in hypothalamic baseline GR levels in both males and females (Males 3α,5α-THP NFSS: 81 ± 4.8% vs. VEH NFSS, Males 3α,5α-THP FSS: 81 ± 6.1% vs. VEH FSS, treatment effect *p* < 0.01. Females 3α,5α-THP NFSS: 47 ± 5% vs. VEH NFSS, Females 3α,5α-THP FSS: 53 ± 7.7% vs. VEH FSS, treatment effect *p* < 0.0001). Significant effects were found using Two-way ANOVA ^####^
*p* < 0.0001, ^##^
*p* < 0.01, ^#^
*p* < 0.05, or following Tukey HSD test **** *p* < 0.0001, ** *p* < 0.01, * *p* < 0.05. Data are represented as % ± SEM vs. VEH no stress. Abbreviations: VEH = rats treated with vehicle; 3α,5α-THP = rats treated with 3α,5α-THP; NRS = non-restraint stress; RS = restraint stress; NFSS = non-forced swim stress; FSS = forced swim stress. All % values are calculated vs. VEH no stress control group.

**Figure 3 biomolecules-12-01134-f003:**
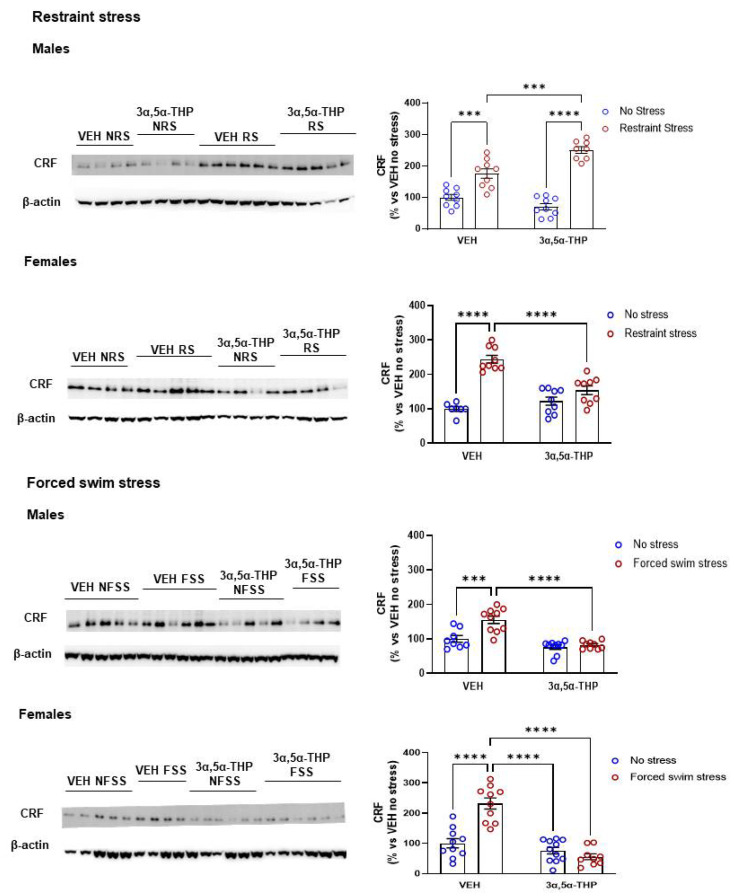
**3α,5α-THP attenuated hypothalamic CRF stress-induced response in females following both stressors and in males following forced swim stress, but failed to reduce the restraint stress-induced increase in CRF in male rats**. As expected, following restraint stress or forced swim stress, hypothalamic CRF levels increased in male and female rats (Males VEH RS: 176 ± 15% vs. VEH NRS, *p* < 0.001; Males VEH FSS: 155 ± 10.7% vs. VEH NFSS, *p* < 0.001. Females VEH RS: 244 ± 11.4% vs. VEH NRS, *p* < 0.0001; Females VEH FSS: 232 ± 18.5% vs. VEH NFSS, *p* < 0.0001). 3α,5α-THP reduced hypothalamic CRF stress-induced in males after forced swim stress (Males 3α,5α-THP NFSS: 76 ± 6.5% vs. VEH NFSS, n.s.; Males 3α,5α-THP FSS: 82 ± 3.7% vs. VEH FSS, *p* < 0.0001), but not after restraint stress (Males 3α,5α-THP NRS: 70 ± 9.9% vs. VEH NRS, n.s.; Males 3α,5α-THP RS: 251 ± 10.5% vs. 3α,5α-THP NRS, *p* < 0.000; Males 3α,5α-THP RS 56 ± 9.6% vs. VEH RS, *p* < 0.001). In female rats, 3α,5α-THP reduced hypothalamic CRF expression following both restraint stress and forced swim stress (Females 3α,5α-THP NRS: 122 ± 12% vs. VEH NRS, n.s.; females 3α,5α-THP RS: 154 ± 12.7 vs. VEH RS %, *p* < 0.0001; Females 3α,5α-THP NFSS: 76 ± 10.9%vs VEH NFSS, n.s.; Females 3α,5α-THP FSS: 56 ± 9.8% vs. VEH FSS, *p* < 0.0001). Significant effects were found using Two-way ANOVA, followed by Tukey HSD test, **** *p* < 0.0001, *** *p* < 0.001. Data are represented as % ± SEM vs. VEH no stress. Abbreviations: VEH = rats treated with vehicle; 3α,5α-THP = rats treated with 3α,5α-THP; NRS = non-restraint stress; RS = restraint stress; NFSS = non-forced swim stress; FSS = forced swim stress. All % values are calculated vs. VEH no stress control group.

**Figure 4 biomolecules-12-01134-f004:**
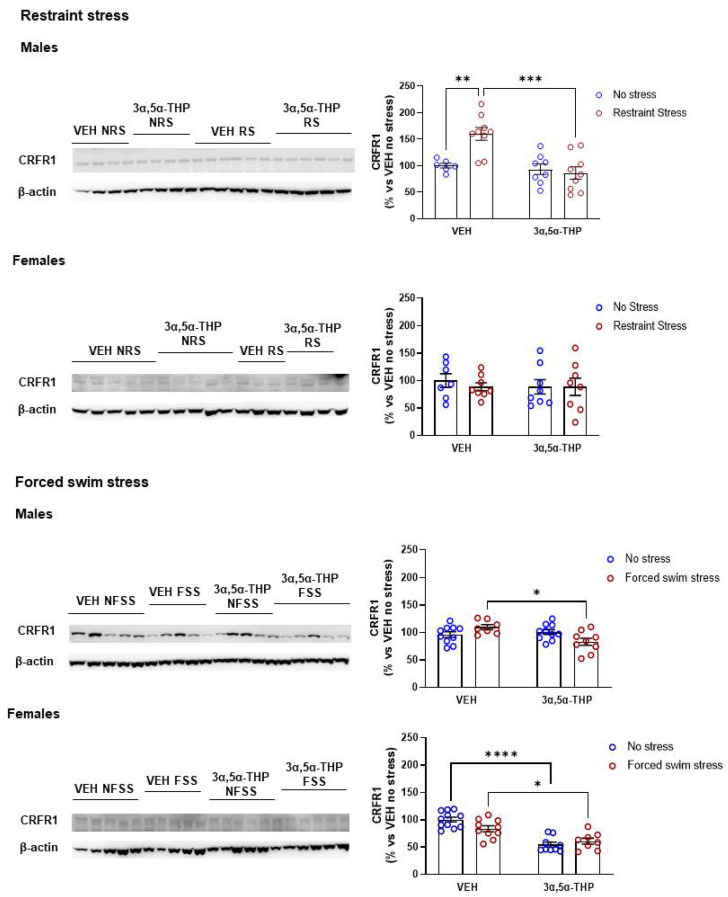
**3α,5α-THP reduced restraint stress-induced hypothalamic CRFR1 increase in male rats.** Hypothalamic CRFR1 levels increased in male rats following restraint stress (Males VEH RS: 160 ± 12% vs. VEH NRS, *p* < 0.01), but not following forced swim stress (Males VEH FSS: 110 ± 4.8% vs. VEH NFSS, n.s.). 3α,5α-THP reduced hypothalamic CRFR1 stress-induced in males after restraint stress (Males 3α,5α-THP NRS: 93 ± 9.8% vs. VEH NRS, n.s.; Males 3α,5α-THP RS: 86 ± 11.6% vs. VEH RS, *p* < 0.001), and CRFR1 levels after forced swim stress (Males 3α,5α-THP NFSS: 101 ± 4.5% vs. VEH NFSS, n.s.; Males 3α,5α-THP FSS: 83 ± 6.5% vs. 3α,5α-THP NFSS, *p* < 0.05). Restraint or forced swim stress did not induce any changes in hypothalamic CRFR1 in female rats (Females VEH RS: 89 ± 7.2%, n.s. vs. VEH NRS; Females VEH FSS: 84 ± 5.4%, n.s. vs. VEH NFSS). However, 3α,5α-THP reduced hypothalamic CRFR1 with and without forced swim stress in female animals (Females 3α,5α-THP NFSS: 55 ± 4.6% vs. VEH NFSS, *p* < 0001; Females 3α,5α-THP FSS:61 ± 5.4% vs. VEH FSS, *p* < 0.05). Significant effects were found using Two-way ANOVA, followed by Tukey HSD test, **** *p* < 0.0001, *** *p* < 0.001, ** *p* < 0.01, * *p* < 0.05. Data are represented as % ± SEM vs. VEH no stress. Abbreviations: VEH = rats treated with vehicle; 3α,5α-THP = rats treated with 3α,5α-THP; NRS = non-restraint stress; RS = restraint stress; NFSS = non-forced swim stress; FSS = forced swim stress. All % values are calculated vs. VEH no stress control group.

**Figure 5 biomolecules-12-01134-f005:**
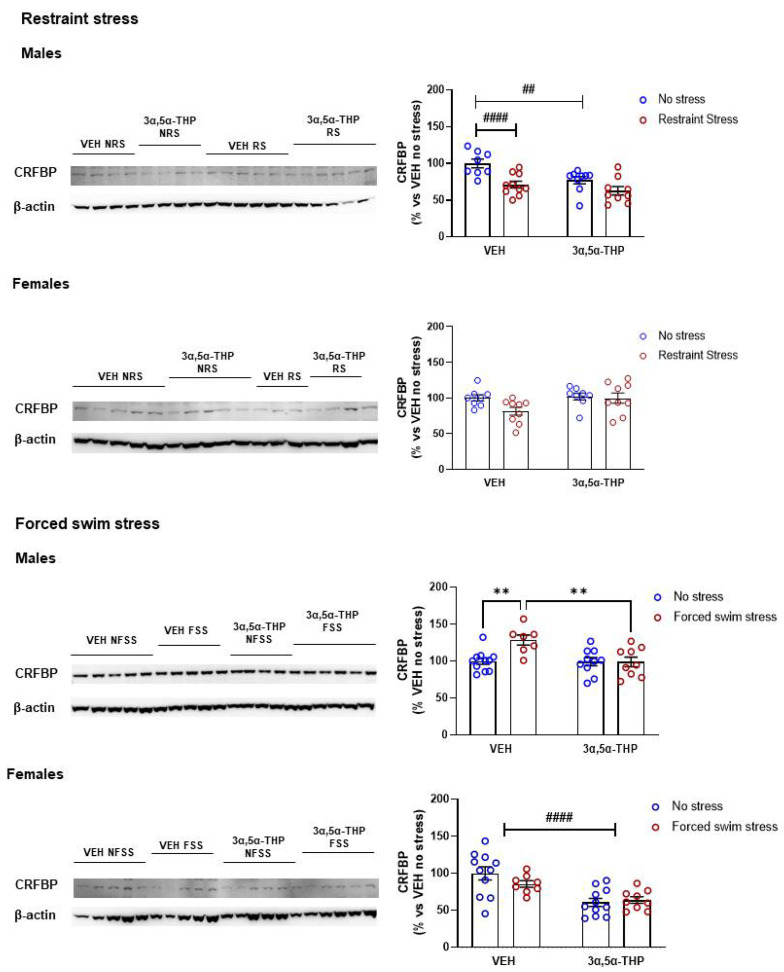
**3α,5α-THP attenuated the forced swim stress-induced hypothalamic CRFBP response in male rats.** In male rats, restraint stress induced a decrease of hypothalamic CRFBP levels (Males VEH RS: 71 ± 4.6% vs. VEH NRS, stress effect *p* < 0.0001), while forced swim stress induced an increase of hypothalamic CRFBP expression (Males VEH FSS: 129 ± 6.8% vs. VEH FSS, *p* < 0.01). 3α,5α-THP treatment did not change the decrease of CRFBP levels in males due to restraint stress (Males 3α,5α-THP NRS: 77 ± 4.9% vs. VEH NRS, treatment effect *p* < 0.01; Males 3α,5α-THP FSS: 99 ± 6.5% vs. VEH FSS, *p* < 0.01). In female rats, restraint stress or 3α,5α-THP administration did not change hypothalamic CRFBP levels (Females VEH RS 81 ± 5.5% vs. VEH NRS, n.s.; Females 3α,5α-THP NRS 102 ± 4.9% vs. VEH NRS, n.s.; Females 3α,5α-THP RS 100 ± 7.3% vs. VEH NRS). 3α,5α-THP treatment resulted in a decrease in CRFBP levels in female animals following forced swim stress (Females 3α,5α-THP NFSS: 61 ± 5.5% vs. VEH NFSS, treatment effect *p* < 0.0001; Females 3α,5α-THP FSS: 64 ± 4.4% vs. VEH FSS, treatment effect *p* < 0.0001). Significant effects were found using Two-way ANOVA ^####^
*p* < 0.0001, ^##^
*p* < 0.01 or following Tukey HSD test ** *p* < 0.01. Data are represented as % ± SEM vs. VEH no stress. Abbreviations: VEH = rats treated with vehicle; 3α,5α-THP = rats treated with 3α,5α-THP; NRS = non-restraint stress; RS = restraint stress; NFSS = non-forced swim stress; FSS = forced swim stress. All % values are calculated vs. VEH no stress control group.

**Figure 6 biomolecules-12-01134-f006:**
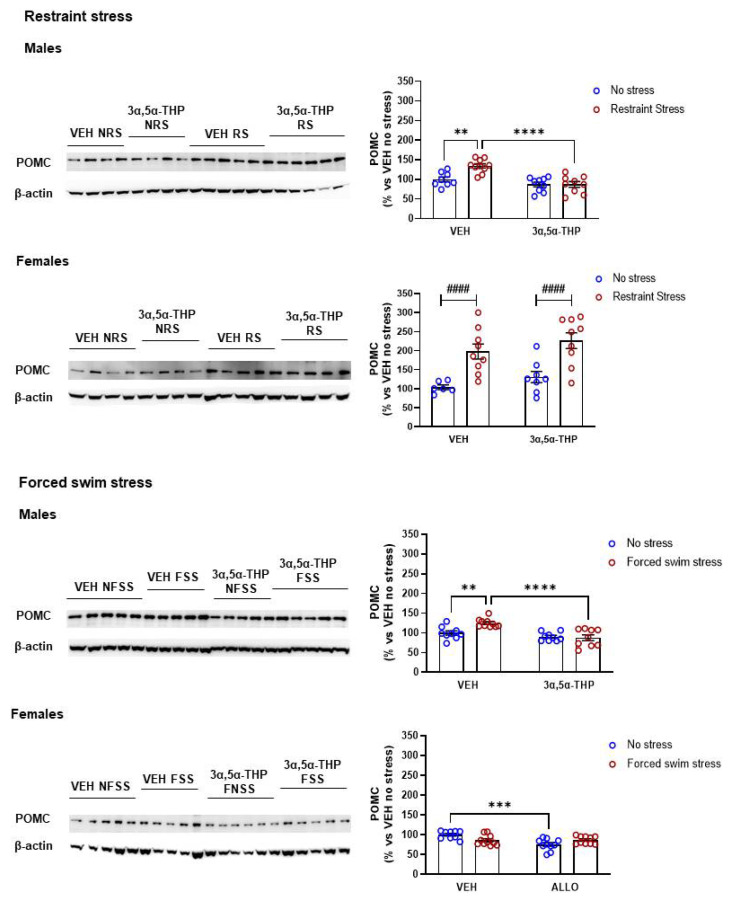
**3α,5α-THP attenuated the restraint stress and forced swim stress-induced hypothalamic POMC increase in male rats, but did not induce any change in female rats.** Male rats exposed to restraint stress or forced swim stress showed an increase in hypothalamic POMC levels (Males VEH RS: 135 ± 5.4% vs. VEH NRS, *p* < 0.01; Males VEH FSS: 125 ± 3.4% vs. VEH NFSS, *p* < 0.01). 3α,5α-THP attenuated the stress-induced increase in hypothalamic POMC levels, following both restraint stress or forced swim stress in males (Males 3α,5α-THP RS: 87 ± 7.3% vs. VEH RS, *p* < 0.0001; Males 3α,5α-THP FSS: 88 ± 7.3%, *p* < 0.0001). Hypothalamic POMC levels increased following restraint stress in female rats (Females VEH RS: 198 ± 20% vs. VEH NRS, stress effect *p* < 0.0001), but 3α,5α-THP did not attenuate this increase (Females 3α,5α-THP RS: 226 ± 21% vs. 3α,5α-THP NRS, stress effect *p* < 0.0001; Females 3α,5α-THP RS: 226 ± 21% vs. VEH RS, n.s.). Forced swim stress failed to increase hypothalamic POMC levels in female rats, but 3α,5α-THP administration decreased POMC levels in the absence of stress (Females 3α,5α-THP NFSS: 75 ± 4.5% vs. VEH NFSS, *p* < 0.001). Significant effects were found using Two-way ANOVA ^####^
*p* < 0.0001 or following Tukey HSD test **** *p* < 0.0001, *** *p* < 0.001, ** *p* < 0.01. Data are represented as % VEH no stress ± SEM. Abbreviations: VEH = rats treated with vehicle; 3α,5α-THP = rats treated with 3α,5α-THP; NRS = non-restraint stress; RS = restraint stress; NFSS = non-forced swim stress; FSS = forced swim stress.

**Table 1 biomolecules-12-01134-t001:** 3α,5α-THP effects on stress-induced response of different markers, following restraint stress or forced swim stress, in (**a**) male and (**b**) female rats.

**(a)**				
	** Males ** **♂**			
	** Restraint Stress **		** Forced Swim Stress **	
** Marker **	** Stress **	** 3α,5α-THP **	** Stress **	** 3α,5α-THP **
** CORT **	+595% **** vs. VNRS	No effect	+266% **** vs. VNRS	No effect
** CRF **	+76 *** vs. VNRS	+150% **** vs. VNRS	+55% vs. VNFSS	−73% **** vs. VFSS
** GRs **	+29% * vs. VNRS	−36% ** vs. VNRS; −37% **** vs. VRS	No effect	−19% * vs. VNFSS
** CRFR1 **	+60% ** vs. VNRS	−67% *** vs. VRS	No effect	−17% * vs. VFSS
** CRFBP **	−29% ** vs. VNRS	−27% * vs. VNRS	+29% ** vs. VNFSS	−30% ** vs. VFSS
** POMC **	+35% ** vs. VNRS	−48% **** vs. VRS	+25% ** vs. VNFSS	−37% **** vs. VFSS
** (b) **				
	** Females ** **♀**			
	** Restraint Stress **		** Forced Swim Stress **	
** Marker **	** Stress **	** 3α,5α-THP **	** Stress **	** 3α,5α-THP **
** CORT **	+295 **** vs. VNRS	−69% ** vs. VRS	+321% ****	No effect
** CRF **	+144% **** vs. VNRS	−90% **** vs. VRS	+132% **** vs. NFSS	−176% **** vs. VFSS
** GRs **	+41% ** vs. VNRS	−13% vs. VRS	No effect	−52% **** vs. VFSS; −47% **** vs. VNFSS
** CRFR1 **	No effect	No effect	No effect	−45% **** vs. VNFSS; −39% * vs. VFSS
** CRFBP **	No effect	No effect	No effect	−39% **** vs. NFSS; −46% * vs. VFSS
** POMC **	+98% ** vs. VNRS	+126% ** vs. VNRS	No effect	−25% *** vs. NFSS

Abbreviations: VNRS = VEH no restraint stress; VRS = VEH restraint stress; VNFSS = VEH no forced swim stress; VFSS = VEH forced swim stress. Significant effect was found using Two-way ANOVA, followed by Tukey HSD test: **** *p* < 0.0001, *** *p* < 0.001, ** *p* < 0.01, * *p* < 0.05. Data are express in % vs. correspondent VEH groups.

## Data Availability

Not applicable.

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
