# Peer review of "(3α,5α)3-Hydroxypregnan-20-one (3α,5α-THP) Regulation of the HPA Axis in the Context of Different Stressors and Sex"

_biomolecules, 2022, doi:10.3390/biom12081134_

Round 1
Reviewer 1 Report
The manuscript described 3a, 5a-THP (a positive allosteric modulator of GABAergic neurotransmission)- mediated regulation of stress-induced alterations of hypothalamic CRF, CRFR1, CRFBP, POMC, and GR expression, and plasma corticosterone levels in adult male and female Sprague Dawley rats. The authors demonstrated that while stressful stimulus (30 min restrain stress or 10 min forced swim test) increased levels of plasma corticosterone in both male and female rats, 3a, 5a-THP treatment attenuated corticosterone response only in female rats exposed to restrain stress and decreased GR expression in males and female rats in a stress stimulus-dependent manner. The authors therefore concluded that 3a; 5a-THP regulates the HPA axis in a stress stimulus and sex-dependent manner.
The purpose, rationale, and experimental design as presented are clear. However, results, discussion, and conclusion are “disconnected”. Accordingly, the discussion and conclusion of the manuscript should be focused on the findings of the study.
Major concerns:
- On page 28, third paragraph the authors surmised that “it is possible that in male rats 3a, 5a-THP is increasing CRF concentration to protect the neurons from neurotoxicity due to increase of glutamate following by stress exposure” Do the authors have evidence indicating that the acute stress stimuli used in the current study results in the release of toxic levels or glutamate? This part of the discussion should be rephrased as it relates to the findings in this study.
- According to the results presented, both restrain stress and 3a, 5a-THP treatment increased CRF expression and these affects are additive or synergistic. However, in page 28 lines 543-547, the authors inferred that “lower CRFBP levels could lead to an increase of CRF activity that 3a, 5a-THP fail to attenuate” Also, based on Fig. 5, restrain stress decreased expression of CRFBP while 3a, 5a-THP treatment had no effect. Please reconcile this inference with the findings presented in Fig. 3 and 5.
- On page 18, lines 588-589, the authors state “3a, 5a-THP reduced levels both in absence and following restraint stress in male rats and the GR increased following restraint stress in females, attenuating the HPA axis activation after stress exposure” However, Fig. 2 shows that there is no statistical significance difference for GR expression following 3a, 5a-THP treatment with no stress and restraint stress; indicating that 3a, 5a-THP treatment had no effect. Please paraphrase or reconcile this statement with the data presented in Fig. 2.
Minor concerns:
- Page 7, line 270; paraphrase to read “It is well known that stress stimulus increases hypothalamic CRF levels, leading to a cascade of events called the stress response.
- Page 7, line 275, Restrain stress increased (instead of amplified).
- Page 9, line 319; paraphrase to read “In contrast” instead of “on the contrary”
- Page 16, line 449; paraphrase to “Acute stress-induced increase of 3a, 5a-THP is considered part of the HPA axis negative feedback mechanisms”
- Page 16, line 452; paraphrase to “genomic pathway by downregulating gene expression”
- Page 16, line 453; paraphrase to “HPA axis activation by reducing corticosterone levels”
- Page 16, line 458; paraphrase to “This paradoxical effect (instead of discrepancy)”
- Page 16, line 465; paraphrase to “But, the contrasting findings (instead of discrepancy)”
- Page 16, line 471; please indicate what is being collected (e.g., tissues, plasma etc).
- Page 16, line 484; paraphrase to, “Upon further evaluation we found that circulating corticosterone”
- Page 16, lines 498-499; paraphrase to “Similarly, exposure to forced swim stress increase circulating corticosterone levels to similar magnitude in both male and female groups. Interestingly, the magnitude of the increase in corticosterone following swim stress exposure”
- Page 17, line 510; paraphrase to “stress response in the hypothalamus in a sex- and stress stimulus-dependent manner.
- Page 17, lines 513-515; paraphrase to “HPA axis by increasing the expression of GR, CRF, and POMC in male and female animals. In contrast, forced swim stress did not produce/induce changes in GRs and CRFR1 in both male and female rats or alter CRFBP and POMC levels in female rats”
- Page 17, line 519; paraphrase to “we observed that 3a; 5a-THP had no effect in female rats but significantly decreased CRF mRNA (but not peptide expression) in male rats. Delete “but this effect was not present in female”
- Page 17, line 527; paraphrase to “These results suggest differences in the regulation of HPA axis by 3a, 5a-THP”
- Page 17, lines 539-540; paraphrase to “force swim stress increased the levels of this protein in male rats. These diverse effects suggest a stress stimulus-dependent HPA axis activation. Delete “dependent of the type of stress”
- Page 17, line 544; to be consistent here and throughout the text, please use CRF levels not concentrations.
- Page 18, line 561; paraphrase to “observed in these disorders”
- Page 18, line 597; paraphrase to “The decrease in 3a, 5a-THP-induced decrease of GR expression was higher in females than in males”
- Page 19, line 606; speculate on how you can prove or demonstrate that 3a, 5a-THP effect is independent of the well known effects of 3a, 5a-THP on GABAergic transmission.
- Page 19, line 612; does 3a, 5a-THP have ability? Please rephrase using better scientific terms.
- Page 20, line 657; the authors concluded that “These results provide a more complete understanding of HPA axis biomarkers” Though the findings are true, they are not exhaustive or a final prove; please paraphrase and expand on future experiments to extend your findings.
- There are several grammatical and typographical errors throughout the manuscript which made the reading and comprehension of the discussion section difficult. The authors may need English editing assistance prior to submitting the revised version of the manuscript.
Reviewer 2 Report
In this very nice study, the authors set out to test whether 3α,5α-THP pre-treatment might have different effects on the HPA axis response to two different stressors. The findings were interesting and showed that 1) CORT increased in response to both stressors in both males and females, validating that the stressors are effective and produce this expected result; 2) that 3α,5α-THP reduced the CORT response to restraint stress (but not forced swim stress ) in females, 3) but not in males regardless of type of stressor; 4) lastly, 3α,5α-THP had sex- and stress type- specific effects on other markers of the stress response, such as hypothalamic CRF release, hypothalamic CRFBP, CRFR1 expression, hypothalamic GR expression, and POMC levels in the hypothalamus.
This is very elegant study, with a thoughtful design that assessed important components of the stress response to test 3α,5α-THP effects on such. The findings are intriguing and raise many interesting questions and hypotheses that can be studied in follow up. Some minor considerations are suggested below in order to clarify/improve select of component of the manuscript.
1. Were 3α,5α-THP levels measured before and/or after the stressors?
2. Figure 1 might be easier to read and interpret if it were to be re-organized to look like the other figures, i.e. by stressor type and then males and females side by side with identical scale on the y-axis.
3. The duration of the swim stress was 10min and the restrain stress 30min, which means that blood and tissue was collected immediately after the restrain stress without any recovery, but 20 minutes after the swim stress stopped. Could some of the differing findings be related to the differences in time course/duration rather than the type of stress?
4. Related to this, it might be helpful if the authors added information and discussion of the half-life of 3α,5α-THP and its relevance to the time course of the two different stressors.
5. Please closely review and correct minor grammatical errors throughout the entire manuscript, e.g. page 2, line 62 “share”, line 63 “differ FROM each other”, page 3, line 113 delete extra “divided”, page 7, line 252 “increase_ in male and..”, page 16, line 457 “SurprisingLY…”, line 500 “InterestingLY…”, etc.
Reviewer 3 Report
Neurosteroids play a crucial role in maintaining and regulating the body’s function, mediating many important physiological processes, such as reproduction, sexual differentiation, ionic and carbohydrate homeostasis and stress responses. In a number of brain structures, these hormones are responsible for diverse neurochemical processes both during development and in adulthood.
It has been a pleasure to evaluate this very timely and elegant research article by Boero and colleagues. Authors show the first time that different levels of HPA axis regulation by 3α,5α-THP depends on the type of stressogenic stimuli and sex. Moreover they also underline the importance of sex as biological variable in the regulation of the HPA axis, given the differences in steroidogenesis and treatment responses previously reported. Importantly, the recent evidence in clinical studies, using the 3α,5α-THP formulation, brexanolone strongly suggests the therapeutic importance of neurosteroids signaling in the brain.
The pharmacological and behavioural methods are in general very well considered and all procedures such as stress paradigms, immunoblotting and biochemical assays were kept the high standard. Appropriate statistical methods were applied with sufficient number of experimental data. The study is perfectly documented and manuscript is distinctly informative. All figures and tables are also well designed and clear. To sum up, this article may be considered as a valuable contribution to the field of experimental neuroscience and applied psychopharmacology. Congratulations for all Authors.
I have got just one minor suggestion for the Authors:
11. Western blot bars and their legends should be a bit enlarged if possible.
